# Integrating Urban Energy Resilience in Strategic Urban Planning: Sustainable Energy and Climate Action Plans and Urban Plans in Three Case Studies in Italy

**Giovanni Tedeschi** 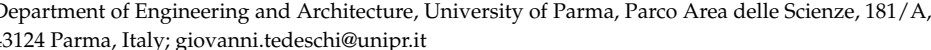

Department of Engineering and Architecture, University of Parma, Parco Area delle Scienze, 181/A, 43124 Parma, Italy; giovanni.tedeschi@unipr.it

**Abstract:** Contemporary cities are facing many challenges, from social and economic issues to the new risks related to the impacts of climate change. Focusing on energy consumptions, and the related GHG emissions, cities are considered not only the main global contributors but also the areas most exposed to risks, because of their density of population and economic activities. Implementing urban planning strategies with the purpose of increasing energy efficiency and resilience overall, is, for all these reasons, considered a top priority. This paper investigates the innovative content related to the energy-efficient and energy-resilient urban planning solutions that have started to be implemented in the cities of the Emilia-Romagna region. Two kinds of planning instruments are therefore analysed: the voluntary Sustainable Energy and Climate Action Plans (SECAPs) and the mandatory General Urban Plans (GUPs), recently approved in several cities of Emilia-Romagna. A comparative analysis of three cities in the Emilia-Romagna region, Bologna, Modena, and Ravenna is proposed, looking at the strategies of their new local city plans and SECAPs with a focus on energy management and planning. The aim is to assess whether the new structure of local city plans and the influence of SECAPs could be useful in implementing such urban-energy resiliency solutions.

**Keywords:** urban planning tools; resilient city; sustainability; SECAPs

## 1. Introduction

Cities and human settlements have been widely acknowledged as crucial focal points in the global endeavour to combat climate change [1]. In this regard, the need for urban adaptation will be a pressing issue for years to come [2]. In the last decades, when the need for global mitigation and adaptation policies and strategies have become urgent, several global agreements, including the United Nations Framework Convention on Climate Change, or UNFCCC [3], the Kyoto Protocol [4] and the 2015 Paris Agreement [5] have emphasized the imperative to curtail climate-altering emissions and to enhance social resilience against the impacts of climate change.

Within this framework, cities are responsible for the production of more than 70 per cent of the world's climate-altering emissions [6], primarily due to their more than 50 per cent population concentration [7]. Consequently, cities are among the territories most susceptible to the impacts of climate change, in terms of economic damage, loss of life and disruption to vital services [8]. The effects and damages of a changing climate on the urban environment have been investigated among different latitudes and urban environments [9,10], with the aim of planning and designing useful response strategies [11].

While mitigation actions often have a top-down structure stemming from major international agreements and are sector-specific, adaptation actions demonstrate greater efficacy when tailored and executed at the local level, with a recognised increasingly important role of regional coordination, support for resource-poor local governments, and strategic backing at the national level [12].

The role of spatial planning in implementing adaptation and mitigation strategies has been debated over the years [13]. Current understanding acknowledges the pivotal role of spatial planning in the endeavour against climate change [14], as it allows for the integration of emission reduction strategies, typically confined to smaller scales such as building energy conservation standards and extending into overarching urban development strategies, thus maximising their impacts on local resilience and development [15,16].

The polysemic concept of resilience has been explored and has declined among different fields of knowledge, from the original ecological background [17] to sustainable farming strategies [18]. Urban resilience is one of the crucial concepts that emerge on the debate on urban planning responses to climate change [19]. Urban resilience is a broad concept that encompass a diversity of approaches, which sometimes overlap, and should be integrated into the wide field of urban sustainability [20]. Urban resilience has been defined as an adaptative ability that allows urban systems to face shocks and chronic stress, while continuing to thrive and evolve [21]. Some contributions stress the importance of urban resilience from an urban management perspective [22]. In this case, urban resilience is seen as the quality of an urban area that helps withstand impacts that could lead to systemic disruption, and which can restore its functions.

This contribution focuses on the less-studied point of view of urban energy resilience [23]. Urban sustainable and resilient energy systems must guarantee availability, accessibility, affordability, and acceptability of energy [24]. To let the urban area adapt, absorb, and recover from any impacts over time, a variety of urban dimensions must be considered: land use, urban morphology, governance, individual behaviour, and socio-demographic aspects. For all these reasons, the integration of energy resilient strategies in urban planning can be considered necessary [25].

From a climate perspective, urban energy consumption is also one of the fundamental contributors to greenhouse gas emissions [26]. Therefore, the ways in which cities manage to develop more efficient and robust energy production and consumption systems holds pivotal significance for mitigation efforts, as well as providing benefits in terms of air quality, health, and quality of life [27]. Conversely, successful adaptation policies on an urban scale possess the potential to significantly diminish the overall costs of impacts and reduce systemic risks [28].

Urban planning tools should play a greater role in addressing these issues [29]. The existing literature reviews suggest that the analysis of the relationship between different planning instruments in relation to climate has not yet been adequately explored, while a better integration of mitigation and adaptation into urban planning is widely advocated [30]. This contribution aims to begin to fill this gap, by exploring, in particular, the interactions on energy issues between voluntary climate action plans and the general urban planning tools. The scope of this research has been narrowed to the climate change strategies and policies of small and medium urban areas, a field that is still not widely explored [31]. To investigate this issue in an area where these topics have been widely discussed and regulated [32], we have chosen as a field of study the medium-sized city of Emilia-Romagna, representative of the Italian and regional urban realities, and characterised by distinct challenges and opportunities in addressing climate change effects [33].

The question that this contribution addresses is whether and how climate action plans, like the Sustainable Energy and Climate Action Plans, can influence, or should influence, the General Urban Plans, in achieving a higher level of energy resilience.

In particular, this article aims to carry out the following: investigate the instruments, encompassing action plans and coordination networks, which cities adopt to respond to climate change, both nationally and internationally (Sections 2.1 and 2.2); present the case selection and the comparative analysis methodology (Section 2.3); explore the relationships between the Sustainable Energy and Climate Action plans (SECAPs) and General Urban Plans (GUPs) in the average Emilia-Romagna city (Section 3); and finally, propose criteria for integrating climate-related considerations into municipal urban planning, with particular reference to energy efficiency and resilience (Section 4).

## 2. Materials and Methods

### 2.1. Policies and Instruments for Tackling Climate Change in the Urban Environment

2.1.1. International Urban Climate Networks

As explained in Section 1, the recognition of the pivotal role urban centres can play in mitigating the multifaceted challenges posed by climate change resonates deeply with key tenets of urban planning theory. The acknowledgment of the role cities can play in the challenges posed by climate change has led to bottom-up engagement and coordination among various local institutions globally [34]. Drawing upon these theoretical underpinnings, the ensuing section of this discourse aims to show the effectiveness of city climate networks in fostering sustainable urban development. Through a series of case studies and examples, this section will demonstrate how collaborative initiatives at the local level can contribute to mitigating the impacts of climate change while promoting resilience and equitable development within urban communities. The ensuing section highlights examples of city climate networks.

One of the most influential instances of city networks centred on environmental issues is Agenda 21, a landmark initiative that catalysed numerous cities worldwide to embrace a Local Agenda 21 framework. This initiative represents a localized implementation of the sustainable development principles outlined by the United Nations following the historic Rio de Janeiro conference in the early 1990s [35]. Grounded in the recognition of the interdependence between ecological health, social equity, and economic prosperity, Agenda 21 aimed to empower municipalities to take proactive measures towards achieving sustainability goals at the local level.

In the subsequent years, a multitude of additional city networks emerged, each embodying a commitment to addressing pressing environmental challenges through collective action. Among these, a notable example is the International Council for Local Environmental Initiatives (ICLEI), established in 1990 [36]. The ICLEI is dedicated to facilitating the integration of climate change considerations into broader local sustainability agendas. Its overarching mission is to empower municipalities worldwide to become catalysts for positive environmental change while simultaneously promoting social equity and economic prosperity.

Founded in 2005, the C40 Cities initiative initially comprised 40 large cities globally. Over the years, this initiative has undergone significant expansion, now encompassing a network of 96 major cities worldwide, with notable participants including Milan and Rome representing Italy's commitment to the cause [37]. The primary objective of the C40 Cities initiative is twofold: to substantially reduce greenhouse gas emissions from member cities by 50% within a decade, while simultaneously bolstering efforts to enhance adaptation measures and air quality policies on more than 582 million citizens worldwide, equivalent to about 36% of the world's GDP.

The Covenant of Mayors initiative, inaugurated by the European Commission in 2008, represents a milestone in the collective effort to combat climate change and advance sustainable development. Rooted in the principle of voluntary cooperation, it is based on a platform for collaboration between signatories such as local and regional authorities and the European Commission, with the aim of achieving and surpassing the climate and energy targets set by the European Union [38].

Through a combination of voluntary commitments, peer learning, and capacity-building initiatives, the Covenant of Mayors empowers local governments to take proactive measures to reduce greenhouse gas emissions, enhance energy efficiency, and promote renewable energy deployment.

The Sustainable Energy and Climate Action Plans (SECAPs) are the fundamental tool of the Covenant of Mayors. These action plans support three main objectives. Firstly, mitigation strategies are carried out, aligning with the European Union's commitment to cutting net greenhouse gas emissions by 55% compared to the 1990 level and to achieve climate neutrality by 2050. The second objective is to improve the adaptation of cities to the negative consequences of climate change through systemic actions, such as a better

data-based knowledge of local vulnerabilities and localized best practices for climate-risk management and prevention. Lastly, the SECAPs aim to reduce energy poverty by granting universal access to safe, clean, and affordable energy for everyone.

The municipality which chooses to join the Covenant elaborates on the SECAP with the support of the Covenant of Mayors' office at the European Commission and the Joint Research Centre [39] for technical issues. Apart from implementing and monitoring actions, subscribing municipalities pledge to share best practices, experiences, and knowledge with the Covenant of Mayors network through institutional cooperation.

Consequently, the SECAPs are important documents for assessing the policies and actions of municipalities that impact energy and resilience. They are composed of three parts: the Baseline Emission Inventory (BEI), which assesses the situation of greenhouse gas emissions; the Risk and Vulnerability Assessment (RVA), concerning the analysis of climate risk in terms of adaptation; and the Plan of actions, which encompass both mitigation and adaptation measures.

The BEI establishes a base year and covers key sectors such as municipal, tertiary, residential, and transport, assessing energy consumption and carbon dioxide equivalent emissions [40]. It is recommended in the guidelines that the focus should be on interventions involving public buildings or services, both for reasons of increased implementation possibilities and to ensure a leading role for the municipal administration in driving change in society and the local economy.

The RVA assesses the hazard, exposure, vulnerability, and resilience aspects of human and material assets threatened by climate change, using either spatial impact models or indicator-based vulnerability analyses. The assessment can be carried out with different levels of detail and depth, depending on the size of the city. Spatial impacts models allow for the appreciation of the variations of the risk levels throughout the area. A simpler methodology, suitable for smaller municipalities with fewer resources, is an indicator-based vulnerability analysis. It starts with an assessment of the qualitative aspects of climate hazards in the territory which lead to point values of vulnerability and exposure.

Following the knowledge framework implemented by the BEI and the RVA, the SECAPs outline a series of actions for achieving specific objectives in emission reduction and increased resilience by 2030.

These actions include short-to-medium-term initiatives, detailed with action sheets specifying descriptions, responsible entities, timelines, affected municipal sectors, the impact focus, expected results, the stakeholder group involved, investment costs, and monitoring indicators.

2.1.2. Examples of Relevant Action Plans

In addition to the study of SECAPs, it was deemed necessary to analyse a selection of climate action plans deemed noteworthy for comprehending current best practices and gathering some insights into energy and resilience in terms of climate planning.

The 2030 Climate Emergency Plan [41] of the city of Barcelona is the outcome of a substantial participatory process engaging numerous associations, together with local institutions and citizens, joining existing climate networks or creating ad hoc aggregations to implement climate change projects in the city.

The plan places emphasis on energy poverty, health risks, and inequalities, influencing the selection of indicators.

Furthermore, it aims at developing a low-carbon city, independent of fossil fuels, and distributing the economic benefits of innovations among the citizens.

The plan also promotes sustainable mobility and the closure of material and energy cycles, with various actions.

The Air and Climate Plan (CAP) of Milano [42] stands out because of the participatory process that involved the municipality and stakeholders. This plan adopts an integrated approach that addresses air pollution in conjunction with mitigation and adaptation measures, an aspect often overlooked or treated separately in sectoral documents. The plan's

strategic actions often affect more than one aspect simultaneously, maximizing synergies and achieving multiple objectives.

In the long term, by 2050 the objectives are the following: compliance with the values set by the WHO Air Quality Guidelines, carbon neutrality, and containment of the local temperature increases to within 2 °C, through urban cooling actions and reduction in the heat island phenomenon.

The SECAP of the city of Mantua [43] envisions a model of distributed energy generation that could improve the relationship between energy, territory, nature, and urban layout.

Beyond its environmental significance, the low-carbon economy is seen also as an opportunity for sustainable economic development and improved quality of life in the territory. This commitment to energy-saving transformation and greater use of renewable energy sources, however, must necessarily be balanced with imperative safeguarding and conservation requirements of the great historical–artistic value of the city centre.

Furthermore, Mantua has a significant industrial presence, which significantly influences the city's emissions trend with its strategic choices. Synergies between these initiatives, such as the district heating network fuelled by waste heat, and urban energy planning can contribute to the comprehensive approach taken by Mantua in addressing climate change.

### 2.1.3. Considerations

The examples examined illustrate how issues related to energy and resilience can be integrated into a developmental framework, creating new opportunities and maximizing the benefits of transformative processes for a broad spectrum of stakeholders. To ensure accurate reception by citizens, the participatory dimension must be taken care of and pursued at all stages of the process, from plan development to implementation and monitoring. The most comprehensive plans, as in the case of Barcelona and Milan, are those that devoted considerable attention to the participatory process in determining the needs to be met and the objectives to be achieved.

The dimension of social equity and inclusiveness might at first seem secondary in a climate plan, but plays a crucial role because it allows for a more precise targeting of measures and a more efficient allocation of resources, and helps to safeguard precisely the most vulnerable elements. In essence, it functions as an adaptation measure, contributing to the overall increase in urban system resilience.

### 2.2. *The Climate Change Mitigation and Adaptation Approach in the Medium-Sized Cities of Emilia-Romagna in Italy*

#### 2.2.1. The Specificity of the Medium-Sized City

Medium-sized cities constitute a significant reality in the European and Italian context. European countries, compared to the rest of the world, exhibit, for historical and geographical reasons, a higher percentage of their population in medium-sized and small cities, with densities lower than in Asian cities but much higher than in US cities [44]. Europe's dense network of medium-sized and small cities tends to be less concentrated around the relatively few large urban agglomerations than they are in other continents.

European institutions have recognized this peculiarity for several years, as evidenced by initiatives like URBAN II, a partnership program with cities conducted between 2000 and 2006. Promoted by the Directorate General for Regional Policies, the program aimed to foster sustainable development in territories facing crises and characterized by this specific urban distribution. European development funds were utilized jointly to address the economic and societal challenges of these regions [45]. More recent research and in-depth programmes strive to better define the specificities and challenges of European medium-sized and small cities through morphological, functional, and administrative analyses [45]. Medium-sized cities actively participate in dedicated European coordination networks, such as the Eurotowns network [46], as well as in forms of competition that enhance and

reward the implementation of sustainable development policies, such as the Green Leaf Award [47].

While large metropolitan concentrations participating in networks like C40 Cities can more readily activate plans and programs, and attract international funding for climate transition policies, medium-sized cities can still play a crucial role in fostering balanced and multi-centred territorial development [48].

This is one of the main reasons of the focus of this contribution on the medium-sized cities of the Emilia-Romagna region in Italy, in three case studies, which share similarities in size and demographic and socio-economic features, and with important differences in the challenges they face in tackling climate change, as explained in Section 2.3.

### 2.2.2. Energy and Resilience References in the Strategies and Urban Planning Laws in the Emilia-Romagna Region

As explained in Section 1, the focus of this contribution are the climate-change and energy-resilience urban strategies of Italian medium-size cities. To make a meaningful selection of case studies, a comparative analysis between spatial governance of the different Italian regions and autonomous provinces was carried out. If we focus only on the instrument of the urban planning law, leaving out strategies, agendas and other documents that are not strictly regulatory for the sake of homogeneity, a diverse situation emerges with some common features [49].

The analysis followed two criteria: a first level that searched for the presence or absence of direct references to climate change mitigation and adaptation, and a second level of analysis relating to related issues such as soil consumption or defence against extreme events.

Emilia-Romagna stands out as one of the most committed Italian regions in integrating mitigation and adaptation solutions into urban planning [50].

This commitment is evident in documents like the Regional Strategy for Climate Change Mitigation and Adaptation [51], whose aim is to make the territory a zero-emission zone and resilient to the impacts of climate change. It follows the signing in 2015 of the Under2 coalition, which commits the region to reducing its emissions by 20 per cent by 2020 compared to 1990, and by 80 per cent by 2050.

A pivotal step in this endeavour is the Regional Law 24/2017, the New Urban Planning Law, prioritizing the regeneration of urbanized territories to enhance urban and building quality, by focusing on energy and resource efficiency, the environmental performance of building materials, and the comfort of buildings [52].

In the enunciation of the general principles and objectives, in Article 1, among the objectives are those of sustainability, equity and competitiveness of the social and economic system, and the fulfilment of the fundamental rights of current and future generations. The soil is indicated as a common good and a non-renewable resource that performs functions and produces ecosystem services, also towards the prevention and mitigation of hydrogeological instability events and climate change mitigation and adaptation strategies. Hence the importance of soil-consumption reduction. The regeneration of urbanised territories is indicated as the main instrument for urban- and building-quality improvement.

In Article 5 of the law, the region assumes the objective of zero soil consumption to be achieved by 2050, and to this end prepares the instruments of territorial and urban planning with a view to maximising the reuse and regeneration of urbanised territory. The reduction in soil consumption also implies, as the law makes explicit, in addition to the regulation of settlement transformations, the designation of areas to re-become permeable. This constitutes a relevant adaptation strategy as it enhances drainage in the case of extreme events, and it mitigates temperature increases in summer.

Furthermore, the law introduces bonus rules for regeneration projects adopting recognized energy-environmental protocols, providing incentives such as discounts on construction taxes and regional contributions.

The main instrument introduced by this law is the General Urban Plan (GUP), the new standard mandatory urban plan. The GUP focuses on energy efficiency and resilience, i.e., the urban organism's ability to adapt to environmental and social challenges and to react positively to emergencies. To achieve these goals, it is crucial to achieve a comprehensive understanding of the morphological, social, economic, climatic, and environmental context grasped in its dynamic dimension.

The GUP presents differences in competences and objectives with respect to previous planning instruments. In the first place, it identifies the perimeter of the urbanized territory, to differentiate areas suitable for regeneration from those subject to soil-consumption restrictions. It also identifies homogeneous parts of the city requiring uniform discipline, focusing on general objectives for improvement without detailing areas for new settlements or transformations. Furthermore, the GUP focuses on the characteristics and requirements of resilience, i.e., on the urban organism's ability to adapt to challenges and emergencies.

In contrast to previous planning, the GUP intends to pursue a greater integration of urban themes within a broader framework of environmental, social, and economic issues and with relevant sustainability policies and actions from national, to supra-regional, to local level.

In conclusion, the GUP is structured to emphasize the strategic dimension over traditional planning, providing indications and objectives within a continuous process of evaluation and flexibility in decision-making. The evolution towards strategy and flexibility is affecting various spheres of town planning, following a debate on the possible advantages, disadvantages, and concrete applicability of such an approach [53].

The relationships between voluntary action plans (SECAPs), as explained in Section 2.1.1 and the mandatory urban plans (GUPs) in three case studies within Emilia-Romagna will be explored to understand the dynamics and methodologies employed in these municipalities.

*2.3. Three Case Studies*

2.3.1. Case Selection Criteria

As explained in Section 2.2.2, the Emilia-Romagna region can be considered as an interesting best practice for investigating the integration of urban energy resilience and urban planning. An initial analysis of the urban areas of the selected region has been carried out, to make an effective choice of case studies. In some cases, the cities have approved the SECAP, but the Local Urban Plans have been approved years before, so no significant references to climate change or urban resilience must be expected. In other cases, both the SECAP and PUG are present, but the city size has been considered too small to be relevant, and is considering in the future to expand the present methodology to international cases.

To make a coherent and expandable choice of case studies, several criteria have been applied. In the first place, medium-sized cities in the Emilia-Romagna region, as depicted in Figure 1, which are also provincial capitals, have been selected. These cities are also representative of diverse territorial and socio-economic contexts: agricultural, industrial, and coastal areas, together with historical heritage city centres are featured in various combinations. Another key requisite is that cities need to have both an approved SECAP and a GUP. It would likewise be useful to choose municipalities that have been engaged for years in initiatives on sustainable development and efforts against climate change, as explained in the following paragraphs.

According to these criteria, the cities of Bologna, Modena and Ravenna have been chosen. Their key features are summarised in Table 1.

**Table 1.** Population and extension of three case studies.

|  | **Bologna** | **Modena** | **Ravenna** |
|---|---|---|---|
| Population | 391,686 | 184,971 | 157,262 |
| Extension (ha) | 14,100 | 18,300 | 65,300 |

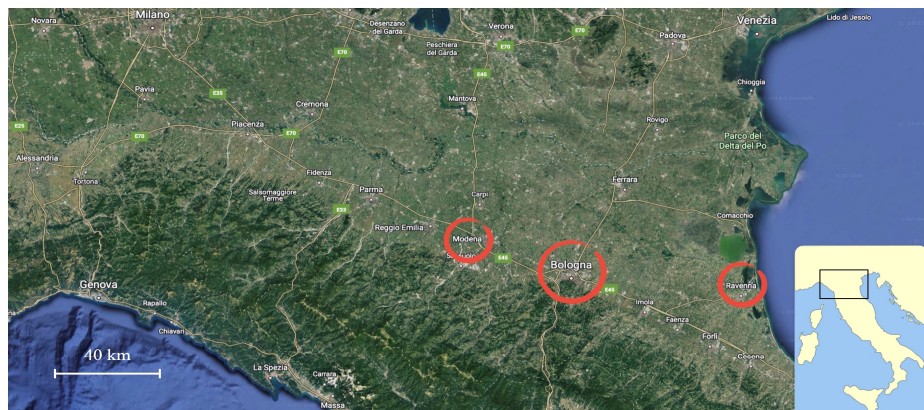

**Figure 1.** Location of the three case studies in the Po Valley, Emilia-Romagna, northern Italy: Modena, Bologna, Ravenna. Elaboration of the author from Google Maps.

Bologna is the capital of the metropolitan cities and of the region Emilia Romagna. It encompasses a population of 391,686 inhabitants and a territorial extension of 141 km² [54]. The metropolitan area covers 3702 km², has a population of 1 million inhabitants, and includes the 55 municipalities of the provincial territory. Bologna is in the southern Po Valley, near the Apennine Mountain range, between the valleys of the Reno River and the Savena stream. The city represents a crucial transportation node for road and railways in northern Italy. It is located at the centre of important east–west and north–south national communication routes. The surrounding area hosts important mechanical, electronic and food industries, as well as cultural institutions.

Modena is the capital city of the province of the same name, with a population of 184,971 inhabitants [55] and an area of 183.19 km². The city is located, like Bologna, in the Po Valley, part of Italy's largest plain. Modena is flanked, but not crossed, by two rivers, the Secchia and the Panaro, and it features an important blue network of canals and watercourses. The territory is considered a crucial hydraulic node for the Po Valley. The city is considered economically one of the major European cities, due to the presence of important food, engineering, and ceramic industries.

Ravenna has a population of 157,262 inhabitants [56], within the second largest municipality in Italy, with a surface area of 653.82 km². The city centre is located 8 km from the Adriatic Sea, to which it is connected by the Candiano Canal. The territory near the coast is characterised by a complex system of beaches, wetlands, basins, floodplains, and natural watercourses, and an elaborate network of artificial canals, the result of a historical process of modification of its morphological and landscape structures by man. It is characterised by the presence of an important port area, connected to the Adriatic north–south national railway lines, and by a globally significant historical, artistic, and architectural heritage.

The selected cities offer a diverse range of characteristics, providing valuable insights into urban planning strategies, mitigation, and adaptation measures in the face of climate change challenges within the Emilia-Romagna region.

### 2.3.2. A Comparative Analysis Method

In the literature, the comparison between different case studies can assume multiple aspects [57]. The use of indicators can be useful when there is the need to assess the performances on resilience and/or sustainability topics, in comparison to known best-practice benchmarks.

In this case, the methodology is based on an in-depth analysis of the planning documents to find the relevant connection between SECAPs and GUPs. A comparative analysis was conducted on the formation of knowledge frameworks and the strategies and actions outlined in the SECAPs and General Urban Plans (GUPs) for the three case studies—Bologna, Modena, and Ravenna, as shown in Figure 2.

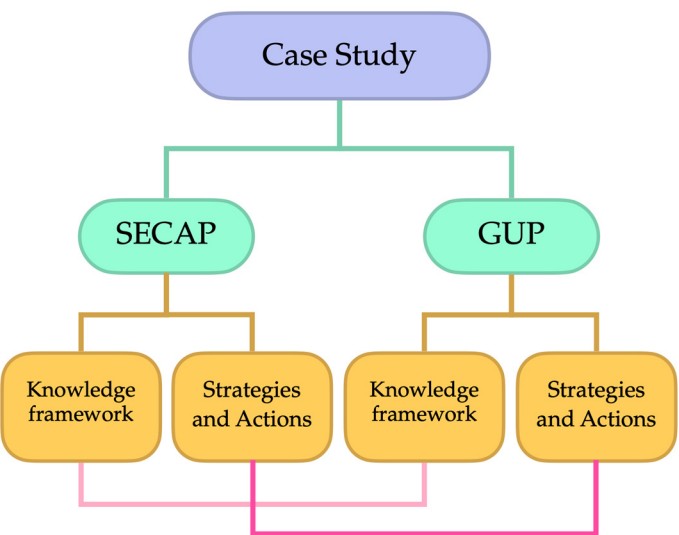

**Figure 2.** Diagram of the method of comparison between the knowledge frameworks and the strategies and actions of the case study's SECAP and GUP.

The comparison aimed to explore the relationships of inclusion, reference, and interference between the various actions of the Action Plans and the articulations of the GUP strategies, particularly focusing on energy management and resilience.

SECAPs and GUPs were compared in relation to the planning and implementation of actions which, directly or indirectly, foster mitigation and adaptation to climate change.

For mitigation, references were sought in the GUPs for strategies involving a reduction in energy consumption, and hence emissions, in the transport, building, production and agricultural sectors. The GUPs, consequently, support the implementation of SECAPs and, more generally, show the contrast with climate change when they are integrated into their strategies and regulations, through the following: the development of pedestrian and bicycle mobility; the support of electric vehicles; the strengthening of the public transport system; the energy efficiency of buildings, of public and private equipment, and of the productive systems; the development of green infrastructure and of an agriculture that improves ecosystem services related to carbon sequestration; the production of energy from renewable sources and the improvement in the efficiency of existing energy production systems; the reduction in waste and waste production; and the raising of citizens' awareness towards more ecological lifestyles.

To make the comparison coherent, it was decided to reorganise the data sources and actions according to the following thematic categories [58]: general strategies, energy efficiency of buildings, public lighting, transport, energy production from renewable sources, waste cycle, green purchasing by public administration, information, awareness, and participation, agriculture, industry, water safety, water resource quality and availability, summer urban comfort, emergency planning and management, and subsidence.

The actions of both plans from every case study have been rearranged through the selected categories and then compared by evaluating the degree of integration.

The following criteria has been applied. Actions concerning the same topic, with a similar level of scale of intervention, like the implementation of renewable energy in residential buildings, have been matched on the same level. Actions concerning the same topic, but with different level of details, were matched by grouping together the smaller ones, e.g., the energy renovation of a particular neighbourhood in the SECAP, to be compared with the more general actions and strategies, e.g., the positive-energy district implementation in the GUP.

The results summarised in the following paragraphs are the result of this iterative work of reference comparison and analysis.

### 3. Results

The comparison of the knowledge frameworks and the strategies and actions between the three case studies allows some considerations. Firstly, the differentiated relationships between the SECAP and GUP in the three case studies emerged. Secondly, different approaches to the energy resilience theme are found, as summarised below.

*3.1. Bologna*

Bologna organises the SECAP, approved in 2021, in macro-chapters with explicit references to the planning actions of the GUP, approved in the same year (Municipal Council Resolution No. 342648 of 26 July 2021). For energy-related issues, both instruments emphasise the importance of promoting the use of national incentives for energy renovation, combined with high-energy performance requirements for urban and building redevelopment and regeneration interventions. Several actions of the SECAP, such as preliminary energy diagnoses and further mapping of energy consumption, support the expansion of the GUP's knowledge framework and the identification of priority areas for intervention, also with reference to publicly owned buildings.

The SECAP refers repeatedly to zero-energy districts (ZEDs) or positive-energy districts (PEDs) as a target for energy efficiency and renewable energy production to be achieved in the areas to be redeveloped. The GUP, in addition to explicitly referring to the energy objectives of the SECAP, specifies high-energy performance requirements in urban and building interventions.

The production of energy from renewable sources (RESs) in the actions of the SECAP and the GUP is addressed through, on the one hand, the prescription of minimum levels of RES coverage in accordance with the general objective of making the city emission neutral. Both instruments emphasise the promotion of neighbourhood energy communities, part of a local and decentralised energy production system, with the aim of achieving 100 per cent coverage from renewable energy sources and providing low-cost energy to combat energy poverty. See Table 2 for an example of a comparison.

**Table 2.** Example of comparative analysis between SECAP and GUP actions in Bologna case study, in three categories considered relevant to energy efficiency and resilience.

| SECAP Actions | Categories | GUP Actions |
|---|---|---|
| Mapping of the energy performance of buildings, actual energy consumption, fuel systems to promote urban transformations towards zero-energy districts, ZED, or energy-producing districts, PED. | Energy efficiency of buildings | Planning measures to encourage renovation and efficiency of the existing building stock. National energy-incentive optimisation. Excellent energy performance for urban requirements for interventions above a certain size. |
| Dissemination and promotion of energy communities by granting public areas to set up large-scale photovoltaic plants to tackle energy poverty. | Energy production from renewable sources | Planning the implementation of energy production plants from renewable sources by creating local distribution networks and energy communities. |
| Increasing logistics efficiency with the creation of proximity logistics spaces and urban freight consolidation centres. Use of zero-emission vehicles. | Transport | Implementation of the Sustainable Logistics Urban Plan to locate and regulate spaces dedicated to freight exchange such as proximity logistics spaces and urban freight consolidation centres. |

*3.2. Modena*

Modena, with a SECAP (2019) that follows the drafting standards of the Covenant of Mayors, incorporates elements of the GUP knowledge framework within the SECAP, taking advantage of the contextual elaboration of the two instruments. The General Urban Plan has been approved by the Municipal Council Resolution No. 46 of 22 June 2023.

The actions in Modena's SECAP relating to the energy efficiency of buildings can be included in the more general actions of the GUP relating to the promotion of energy efficiency in public buildings and to the regeneration discipline for transformations. In the SECAP there are several specific actions concerning the redevelopment of relevant buildings, such as the former AMCM, which are referred as significant regeneration operations to be completed within the GUP framework.

Renewable energy production occupies a considerable section of the SECAP, with four actions relating to the enhancement of photovoltaic production in municipal buildings and the promotion of incentives and energy communities for the private sector. In the GUP the topic is not dealt with at the strategy level, except in an indirect way when talking about energy efficiency in buildings and in an action relating to agriculture in which the creation of photovoltaic parks is promoted, to decrease energy consumption. See Table 3 for an example of a comparison.

**Table 3.** Example of comparative analysis between SECAP and GUP actions in Modena case study, in three categories considered relevant to energy efficiency and resilience.

| SECAP Actions | Categories | GUP Actions |
|---|---|---|
| Palazzo Ducale di Modena energy efficiency project. | Energy efficiency of buildings | Regeneration discipline for transformations to foster widespread energy efficiency, seismic safety, and urban comfort, including through complex urban projects. |
| Regeneration of former cattle market area with the realisation of new office buildings with high energy performance. | | |
| HPE-COXA renovation project. High-performance energy and environmental company buildings with solutions such as green roofs, rainwater harvesting, and photovoltaic panels. | Industry | Measures supporting ecological qualification of production facilities. Improvement in the energy and environmental performance of production sites also through conversion into Eco Industrial Districts (APEA). |
| Citizenship-awareness initiatives to promote sustainable mobility. | Information, awareness, and participation | No specific awareness actions in the Modena GUP. |
| Working group, fostered by the local university, with the aim of developing initiatives to raise awareness of sustainability, energy and mobility issues for staff and students. | | |

*3.3. Ravenna*

Ravenna presents the most complex and articulated of the three case studies, where the GUP (assumed by the Municipal Council Resolution No. 14 of 14 January 2022) deepens in a spatial and strategic way the actions which, in the SECAP (approved in the end of 2020), are treated on a more general level.

The topic of energy efficiency of buildings is present in the SECAP and in the GUP, as in the other two case studies. Only the SECAP emphasises the importance of promoting the use of national incentives for energy requalification to promote energy improvements, while the GUP imposes higher performance requirements than the national standards for urban and building regeneration interventions, with the aim of enhancing buildings' energy efficiency and the urban-energy metabolism overall.

Only the GUP promotes and regulates the energy qualification of industrial and tertiary areas, including tourist facilities on the coast, in an overall design of environmental and energy improvement.

The production of energy from renewable sources (RESs) is addressed differently in the SECAP and the GUP. In the GUP, there is a general reference to innovating the energy cycle, while in the SECAP the state of installation of RES plants in the territory and the development objectives foreseen in the following years are detailed, as well as projects such as the experimental wind turbines in the passenger terminal at the port, or the installation of photovoltaic plants in schools and on public residential buildings. Only the GUP refers to energy communities, while the SECAP proposes to develop the use of RESs more by exploiting the possibilities of the most recent regulations. See Table 4 for an example of a comparison.

**Table 4.** Example of comparative analysis between SECAP and GUP actions in Ravenna case study, in three categories considered relevant to energy efficiency and resilience.

| SECAP Actions | Categories | GUP Actions |
|---|---|---|
| Mapping of the energy performance of buildings, actual energy consumption, fuel systems to promote urban transformations towards zero-energy districts, ZEDs, or energy-producing districts, PEDs. | Energy efficiency of buildings | Planning measures to encourage renovation and efficiency of the existing building stock. National energy-incentive optimisation. Excellent energy performance for urban requirements for interventions above a certain size. |
| Expected increase in photovoltaics due to national energy programs such as the 110% Superbonus and the construction of energy communities. | Energy production from renewable sources | Requirements to include an emission balance in relevant agricultural transformations. Promotion of agricultural solar parks with renewable roofing. |
| Information and awareness-raising campaigns on climate change and energy issues. | Information, awareness, and participation | Fostering of green communities for waste and energy management, with citizens' involvement. |

### 3.4. Summary of Results

The following Table 5 is a summary assessment of the integration of the two instruments in the three case studies, following the categorisation of the analysis and including both knowledge frameworks and strategies and actions. The assessment ranges from excellent integration, when there is full correspondence, to poor, when the sources or actions between the two instruments do not coincide or when in one of the two instruments the topic is not addressed. The full categories of the study are reported, with a highlighting of the integration of energy-related issues.

**Table 5.** Summary of the integration of SECAPs and GUPs in the three case studies. In **bold** the features considered more connected to the energy-related issues.

| Categories | SECAP–GUP Integration | | |
|---|---|---|---|
| | Bologna | Modena | Ravenna |
| **General strategies** | **Excellent** | **Good** | **Very Good** |
| **Energy efficiency of buildings** | **Excellent** | **Very Good** | **Very Good** |
| **Street lighting** | **Very Good** | **Poor** | **Good** |
| **Transport** | **Excellent** | **Very Good** | **Very Good** |
| **Energy production from renewable sources** | **Very Good** | **Good** | **Very Good** |
| Waste cycle | Good | Poor | Good |
| **Information, awareness, and participation** | **Poor** | **Poor** | **Good** |
| Agriculture | Good | Very Good | Poor |
| **Industry** | **Very Good** | **Very Good** | **Poor** |
| Hydraulic Safety | Excellent | Excellent | Excellent |
| Water quality and availability | Very Good | Excellent | Good |
| Urban heat comfort | Excellent | Very Good | Excellent |
| Emergency planning and management | Very Good | Poor | Excellent |
| Subsidence | Not treated | Poor | Excellent |

## 4. Discussion

This contribution delves into the integration of climate change challenges into planning practice, particularly in relation to energy issues.

First, a survey was conducted on the instruments, including action plans and co-ordination networks, which cities adopt to respond to climate change, both nationally and internationally.

Secondly, the relationships between Sustainable Energy and Climate Action Plans (SECAPs) and General Urban Plans (GUPs) in the average Emilian city were explored.

From the analysis of the relationships between the SECAP and the GUP, this contribution tries to address a relevant question: how much, and how, urban resilience, and in particular energy resiliency, can be integrated within the planning instruments.

To achieve this goal, three cases of average cities in the Emilia-Romagna region were studied, which compared to the international and national review of cases can be considered overall examples of good practices. The combination of the presence of a recent regional law with a strong focus on the issues and the presence of cities that have already drawn up their urban plans based on this new approach has made the Emilia-Romagna region an interesting field for identifying urban planning trends related to climate change.

Optimal urban planning for climate change, however, requires a plurality of instruments acting in synergy, to grasp the multi-scalar and multi-functional implications that climate impacts cause in an already complex organism such as the city.

While waiting for the effects of General Urban Plans (GUPs) to unfold on the territory, several considerations can already be drawn from the study of the adopted documents, which may be supplemented in the future with the acquisition of new data and experiences.

These suggestions can be categorized into two key domains: firstly, a structural and content complementarity, which considers ways in which the two plans can complement each other in terms of structure and content organization. Secondly, the procedural and organizational improvements are considered, to refine the drafting and management processes, especially within local government settings.

### 4.1. Structural and Content Complementarity

This analysis suggests that greater integration between Sustainable Energy and Climate Action Plans (SECAPs) and GUPs could positively impact their intended objectives.

GUPs and SECAPs should, therefore, not be considered as separate and independent instruments, as neither of them alone can fully address the complex planning required for energy and resilience issues.

Consequently, it is recommended that greater integration between the GUPs and SECAPs be pursued to optimize synergies and effectiveness while minimizing redundancies and procedural overlaps.

Drawing from the analysis and case studies, the proposed integration criteria between General Urban Plans and PAESCs are provided, intended as guidance for cities in the process of approving new urban planning instruments in Emilia-Romagna.

The SECAP and the GUP should maintain the roles and scopes within which they are more effective. The GUP should remain, given the vastness of its scope and scale of intervention, the potentially most incisive instrument in the transformation of the urban territory and its functions for achieving the transition to carbon neutrality and preparing the city for the impacts of the present and future climate.

The SECAP maintains its original function as a stimulus and solicitation for administrations to implement mitigation and adaptation actions. The presence of innovative actions, especially those of urban relevance, can be the lever through which, at a political-decision-making level, transformative elements can be introduced within the GUP, adding greater value to innovations that may have limited impact if solely confined to the SECAP.

For example, some mitigation measures, such as the purchase of certified electricity from renewable sources or the detailed regulation of the types of vehicles that can circulate according to their level of pollution, should be addressed in the SECAP and not in the GUP. Energy renovation of buildings, likewise, is better applied to singular buildings in the SECAP, while in the GUP it can be implemented in regeneration interventions on a larger scale.

*4.2. Procedural and Organizational Improvements*

The analysis shows how a joint elaboration, in the beginning phase, and a parallel updating of the two instruments to verify their synergies and coherence, could be desirable. The greatest integrations have developed in cases where the SECAP and GUP were drafted together.

This suggests that it would be advantageous, if not imperative, to consider reorganizing the offices responsible for managing different planning instruments, or at the very least, establishing collaborative working groups.

Such collaboration would enable seamless coordination, particularly when dealing with plans developed at different times. By avoiding redundancy, for instance, in duplicating the knowledge framework established for the initial instrument, subsequent planning processes could benefit from the prior analyses without starting anew, potentially from disparate sources. This discrepancy is evident in some instances of comparing knowledge frameworks, where the sources and databases pertaining to the same topic differ between SECAPs and GUPs. Achieving effective integration on energy resilience between these instruments may necessitate administrative restructuring, aligning the competencies and procedures of environmental and mobility offices and those involved in urban planning.

Similarly, the knowledge framework could benefit from a better integration of the SECAP and the GUP, where each instrument can draw useful information from the other.

In the practical realm, the diverse development trajectories pursued by different offices or entities external to the municipality often hinder the attainment of seamless integration. Consequently, there is a looming risk of duplicating analyses that are not directly comparable. Enhanced process integration is essential to mitigate this risk, enabling the efficient establishment of a coherent knowledge framework devoid of redundancies or information gaps.

Furthermore, the development process of the SECAPs and the GUPs should carefully consider the participatory process, which is quite relevant in the examples reviewed. Participation should be strengthened and integrated between the SECAP and GUP, as the results showed, to better grasp in advance problems and conflicts that might arise. It might therefore be appropriate to tend towards a single participatory process, albeit declined in different phases and with different stakeholders, to make the most of the acquisitions

of ideas, needs, suggestions, and directions that might emerge and to communicate to citizens and stakeholders the sense of participating in a coherent and structured process. A strong and experienced participatory process could help bridge the gap between planning strategies at the municipal level and the diverse sets of actions that SECAPs usually envision for organisations, communities, and associations.

*4.3. Limits of the Research and Possible Further Developments*

The proposed criteria for integrating climate considerations into planning could be further examined to assess their applicability across various regional planning contexts, in Italy and potentially in other countries.

Additionally, research could focus on municipal planning structures, aiming to identify barriers and organizational potentials through comparison with international practices, thus proposing improvements in decision-making processes.

Further exploration could delve into the role of monitoring within planning instruments, identifying synergies and additional indicators for effective long-term objective monitoring.

Investigating the impact of other climate change mitigation instruments, such as local climate transition strategies and city climate plans not affiliated with the Covenant of Mayors, could also provide valuable insights into the strengths and weaknesses of the SECAP tool.

This comparative analysis could offer recommendations on essential tools and methodologies for achieving optimal mitigation and adaptation outcomes in typical Italian municipalities, to be compared with similar cases of medium-sized cities at the international level.

**Funding:** This research received no external funding.

**Data Availability Statement:** Data are contained within the article.

**Conflicts of Interest:** The author declares no conflict of interest.

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
