# Peer review of "Integrating Urban Energy Resilience in Strategic Urban Planning: Sustainable Energy and Climate Action Plans and Urban Plans in Three Case Studies in Italy"

_land, doi:10.3390/land13040450_

Round 1

Reviewer 1 Report

Comments and Suggestions for Authors

1. The paragraphs in the introduction section are too short. Please add more information and integrate them.

2. In the beginning of the introduction, you can provide more background on today’s urban development.

3. The literature review in the introduction is quite weak. You should provide a literature review and highlight the research gaps. Specifically, the following 3 aspects should be strengthened. 1) The research progress of resilience, including the concept, role, connotation, etc. 2) The development and current situation of urban planning and urban resilience. 3) The interactions between government policy and resilience. I recommend the following papers to you, and I hope you can read and cite them: 1) Urban resilience for urban sustainability: Concepts, dimensions, and perspectives. 2) Temporal and Spatial Effects of Heavy Metal-Contaminated Cultivated Land Treatment on Agricultural Development Resilience.

4. Also, the paragraphs in the section 2 are too short. I think each paragraph should consist of at least eight lines to ensure coherence and cohesion.

5. Why do you choose these cases for comparison? Please provide more evidence.

6. In the discussion section, I hope the readers use subheadings.

7. Also, please add more information to the discussions. Firstly, please compare the research methodology and conclusions between this study and existing studies. Secondly, please highlight the contributions and limitations of your study.

8. In section 2.3.2, the sentences should be supported by references.

9. Lines 343-360, the format is not good. I suggest the authors to read more papers to learn how to present this section.

10. In sections 3.1, 3.2, 3.3, the introductory part is so limited, and I believe a comprehensive review should be provided.

11. The significance of the research should be highlighted in the introduction and discussion.

Reviewer 2 Report

Comments and Suggestions for Authors

Thank you for selecting a topic that is very relevant today, namely climate change, energy, and environmental consciousness in urban context. I also recognize your expertise on three medium-sized cities that have important regional role. However, there are three major challenges in the manuscript: first, the conceptual background references are almost completely outdated being 10-20 years old while scientific discussion has progressed substantially in the last 10 years; second, there is not clear methodological framework how these cases were analyzed; third, there is not clear conceptual and empirical reasoning how the empirical findings would bring international scientific state-of-the-art forward with this manuscript.  

Reviewer 3 Report

Comments and Suggestions for Authors

Lines 64-66 I'd like to know why you need to cite these literature numbers 15 and 16. I think you can review these two in the previous paragraphs, or the next section (2.1).

Lines 68-184 Section 2.1 needs more case discussion on the international level. I see you only reviewed the case of Barcelona. Please add some more cases to yield in more robust theoretical framework.

Comments on the Quality of English Language

I still found some errors like redundancy and incomplete sentences, for example in lines 100-102. Please have this manuscript proofread and edited by your peers or a professional English editor.

Round 2

Reviewer 1 Report

Comments and Suggestions for Authors

The manuscript has been improved a lot. Several minor suggestions:

1. Please attempt to avoid the format of bulletin points throughout the text.

2. In the discussion section, you can consolidate these sections to improve the coherence. Some subsections are too short and should not be treated as separate subsection.

3. In some sections, you can add figures to better present your ideas.
